# Fetal Undernutrition Induces Resistance Artery Remodeling and Stiffness in Male and Female Rats Independent of Hypertension

**DOI:** 10.3390/biomedicines8100424

**Published:** 2020-10-16

**Authors:** Perla Y. Gutiérrez-Arzapalo, Pilar Rodríguez-Rodríguez, David Ramiro-Cortijo, Marta Gil-Ortega, Beatriz Somoza, Ángel Luis López de Pablo, Maria del Carmen González, Silvia M. Arribas

**Affiliations:** 1Center of Research and Teaching in Health Sciences (CIDOCS), Universidad Autonoma de Sinaloa, Av. Cedros y calle Sauces s/n, Culiacán 80010, Sinaloa, Mexico; perla.gutierrez@uas.edu.mx; 2Department of Physiology, Faculty of Medicine, Universidad Autonoma de Madrid, C/Arzobispo Morcillo 2, 28029 Madrid, Spain; pilar.rodriguezr@uam.es (P.R.-R.); dramiro@bidmc.harvard.edu (D.R.-C.); angel.lopezdepablo@uam.es (Á.L.L.d.P.); m.c.gonzalez@uam.es (M.d.C.G.); 3Department of Medicine, Beth Israel Deaconess Medical Center, Harvard Medical School, 330 Brookline Ave., Boston, MA 02215, USA; 4Department of Pharmaceutical and Health Sciences, Faculty of Pharmacy, Universidad CEU-San Pablo, C/Julián Romea, 23, 28003 Madrid, Spain; mgortega@ceu.es (M.G.-O.); bsomoza.fcex@ceu.es (B.S.)

**Keywords:** fetal programming, hypertension, resistance arteries, metalloproteases, vascular mechanics, vascular remodeling, sexual dimorphism

## Abstract

Fetal undernutrition programs hypertension and cardiovascular diseases, and resistance artery remodeling may be a contributing factor. We aimed to assess if fetal undernutrition induces resistance artery remodeling and the relationship with hypertension. Sprague–Dawley dams were fed ad libitum (Control) or with 50% of control intake between days 11 and 21 of gestation (maternal undernutrition, MUN). In six-month-old male and female offspring we assessed blood pressure (anesthetized and tail-cuff); mesenteric resistance artery (MRA) structure and mechanics (pressure myography), cellular and internal elastic lamina (IEL) organization (confocal microscopy) and plasma MMP-2 and MMP-9 activity (zymography). Systolic blood pressure (SBP, tail-cuff) and plasma MMP activity were assessed in 18-month-old rats. At the age of six months MUN males exhibited significantly higher blood pressure (anesthetized or tail-cuff) and plasma MMP-9 activity, while MUN females did not exhibit significant differences, compared to sex-matched controls. MRA from 6-month-old MUN males and females showed a smaller diameter, reduced adventitial, smooth muscle cell density and IEL fenestra area, and a leftward shift of stress-strain curves. At the age of eighteen months SBP and MMP-9 activity were higher in both MUN males and females, compared to sex-matched controls. These data suggest that fetal undernutrition induces MRA inward eutrophic remodeling and stiffness in both sexes, independent of blood pressure level. Resistance artery structural and mechanical alterations can participate in the development of hypertension in aged females and may contribute to adverse cardiovascular events associated with low birth weight in both sexes.

## 1. Introduction

Fetal life is a key developmental period and a determinant of future health. Barker’s hypothesis states that undernutrition during critical periods of gestation leads to inadequate fetal growth and programs adult disease [1]. The fetus has high plasticity and responds to suboptimal environments by adapting its growth to ensure immediate survival. However, initially adaptive changes may become maladaptive later on, being the origin of disease. Over the last decades, epidemiological and experimental data have provided evidence of this hypothesis and the link between low birth weight (which is a proxy for impaired growth during fetal life), and development of cardiovascular diseases (CVDs). In addition to malnutrition, other stress factors such as hypoxia, placental insufficiency or exposure to toxic substances, can also program the fetus, increasing the risk of hypertension, coronary heart disease and metabolic diseases [2,3]. 

Hypertension development has been consistently associated with suboptimal intrauterine growth, demonstrated in humans [4] and confirmed in animals exposed to various stress factors during fetal life [2]. Abnormal cardiovascular development, stiffening and remodeling are possible mechanisms, which can explain this relationship [3]. For example, fetal stress factors can affect elastin deposition, which later on may compromise vascular elasticity [5]. There is evidence of reduced aortic lumen and wall thickening in individuals with low birth weight, alterations that, later on, can decrease large artery compliance [6,7]. However, the role of vascular remodeling in hypertension induced by fetal programming remains controversial [3]. In aorta from one-month-old rats exposed to fetal undernutrition, we have reported increased wall thickness, but better compliance, and in adult age, no alterations in mechanical behavior were detected [8]. In addition to elastic artery stiffening, resistance artery narrowing and subsequent increase in total peripheral resistance is another key mechanism associated with hypertension [9,10]. Resistance artery remodeling is not only relevant for hypertension development, but also participates in organ damage. In fact, small artery narrowing and stiffening are considered of prognostic value for cardiovascular adverse events [10,11]. Exposure to undernutrition in fetal life is associated with a high risk of coronary heart disease and renal damage [1], and it is possible that resistance artery narrowing and stiffening may be a contributing factor. 

Pathological vascular remodeling is associated with alterations in extracellular matrix (ECM) degradation, and metalloproteases (MMPs)—key enzymes responsible for ECM turnover—have been proposed to play a role in this process [12]. Besides, there is evidence of an association between low birth weight and variations in MMP expression and activity. In a rat model of fetal undernutrition, increased vascular MMP expression has been described at birth [13]. Individuals born with low birth weight also exhibit elevated plasma levels of MMP-2 and MMP-9 in childhood, which correlated with blood pressure and vascular dysfunction [14]. These data suggest an activation of MMPs by fetal stress. On the other hand, there is also evidence that MMP activation may be induced by sustained hypertension, as shown by increased serum concentrations of several MMPs in hypertensive patients and rats, which are reduced upon antihypertensive treatments [15].

Males seem to be more susceptible to adverse fetal environments than females, and the majority of studies, including ours, evidence that female rats do not develop hypertension or develop milder forms in adult age [2,8,16,17]. Based on these data, the impact of adverse intrauterine environments on females is not completely accepted and is sometimes neglected. It has been proposed that females are partially protected from adverse intrauterine environments due to a better adaptation of the placenta, and during adult life due to the influence of sex hormones [18]. However, it is possible that this protective effect is lost in ageing. In fact, we have demonstrated similar cardiac dysfunction in aged male and female rats exposed to fetal undernutrition [19]. 

The aim of the present work was to assess, in a rat model of maternal undernutrition during gestation (MUN), sex differences in the development of mesenteric resistance artery (MRA) remodeling and mechanical function. We have also evaluated the relationship between remodeling, hypertension and MMP. We observed that male and female offspring from MUN rats develop MRA inward eutrophic remodeling and stiffness in adult age, independent on their level of blood pressure. Males develop hypertension in adult age and females in ageing. MRA remodeling can participate in the later development of hypertension in females and can contribute to the cardiac dysfunction previously reported in MUN rats in ageing. Plasma MMP activation is likely a consequence of hypertension, but we cannot discard an activation in the vascular wall, which could contribute to the observed structural and mechanical alterations. 

## 2. Experimental Section

### 2.1. Maternal Undernutrition (MUN) Model and Experimental Design

Experiments were performed on Sprague–Dawley rats from the colony maintained in the Animal House at Universidad Autónoma de Madrid (ES-28079-0000097). All experimental procedures conformed to the Guidelines for the Care and Use of Laboratory Animals (NIH publication N^o^. 85-23, revised in 1996), Spanish legislation (RD 1201/2005), and follow the rules of the Declaration of Helsinki. The experiments were approved by the Ethics Review Board of Universidad Autónoma de Madrid (Ref. CEI-UAM 96-1776-A286; approval date: 20 December 2018) and by the Ethics Board of the Regional Environment Committee of Comunidad Autónoma de Madrid (RD 53/2013; Ref. PROEX 04/19; approval date: 20 March 2019). 

Specialized staff regularly monitored the welfare of the animals and certified that they were free from pathogens, which may interfere with the experiments. The animals were kept with a 12 h/12 h light/dark photoperiod and were maintained at constant temperature (22 °C) and relative humidity (40%). 

A model of fetal programming induced by maternal undernutrition during gestation (MUN) was used, as previously described [17]. Briefly, the day sperm was observed in the vaginal smear was considered first day of gestation. Then, the pregnant rats were allocated to one of the two experimental groups (Control or MUN). All the dams were fed with breeding diet (55% carbohydrates, 22% protein, 4.4% fat, and 4.1% fiber; Euro Rodent breeding Diet 22; 5LF5, Labdiet; Madrid, Spain) and drinking water was provided ad libitum throughout the study. Control rats were fed ad libitum throughout gestation and lactation. MUN rats were fed ad libitum during the first ten days of gestation. From day 11 to delivery, food was restricted to 50% of the averaged Control daily intake, returning to free access to the diet during lactation. 

After birth, the pups were sexed and weighed and the litter was standardized to 12 rats, 6 males and 6 females if possible (litters with less than 12 individuals were not used). Experiments were conducted in male and female offspring at two age points, adult rats (6 months old) and aged rats (18–20 months old), using offspring from different dams (minimum 3 different litters). On the day of the experiment, after blood pressure measurements, the rats were euthanized by overdose of anesthesia and the blood was collected in chilled heparin-coated polypropylene tubes. Blood was centrifuged at 800g for 20 min at 4 °C to obtain plasma. 

In 6-month-old rats the mesenteric bed was excised, placed on chilled zero-calcium Krebs–Henseleit solution (KHS: 115 mM NaCl, 4.6 mM KCl, 25 mM NaHCO_3_, 1.2 mM KH_2_PO_4_, 1.2 mM MgSO_4_, 10 mM EGTA, 11 mM glucose) and 3rd order branches (mesenteric resistance artery, MRA) were carefully dissected. 

### 2.2. Hemodynamic Parameters

Hemodynamic parameters were assessed in 6- and 18-month-old rats by two protocols, intra-arterial measurements under anesthesia, and in awake rats by tail-cuff plethysmography. 

#### 2.2.1. Intra-Arterial Measurements

The 6-month-old rats were anesthetized with 37.5 mg/kg Ketamine hydrochloride and 0.25 mg/kg Medetomidine hydrochloride i.p. A catheter filled with 0.9% saline containing 1% heparin was inserted through the iliac artery and passed into the distal abdominal aorta. The catheter was connected to a pressure transducer (Statham, Harvard Apparatus, Holliston, MA, USA) and to a PowerLab data acquisition system/8SP (ADInstruments, Madrid; Spain) and pressure wave was continuously recorded for 45 min. The diastolic and systolic blood pressure (DBP, SBP; mm Hg) and heart rate (HR; beats/min) were analyzed. Quantification was performed averaging the data in the pressure wave trace for approximately 1 min during the last part of the recording period [8].

#### 2.2.2. Tail-Cuff Plethysmography

SBP was measured non-invasively using a tail-cuff coupled to a pressure acquisition system (CIBERTEC Niprem 645, Madrid; Spain), as previously described [20]. Briefly, the rats were first placed in a chamber at 37 °C for 10–15 min to induce vasodilatation. Thereafter, they were placed inside a soft support in the darkness and the pressure cuff was placed around the tail. The cuff was inflated to 150 mmHg; thereafter, several pressure inflate-deflate cycles were performed, and data were registered. The experiments were performed always by the same investigator. The measurements were performed during 3 consecutive days. Data from day one were not used for quantification, since on the first day the rats were getting used to the procedure and readings may not be accurate.

### 2.3. Pressure Myography

MRA segments were mounted on a pressure myograph (Danish Myotech P100, J.P. Trading, Hinnerup; Denmark), as previously described [21]. Briefly, the MRA segment was placed in an organ bath containing zero-calcium KHS, maintained at 37 °C and bubbled with a mixture of 95% O_2_ and 5% CO_2_, to maintain a pH between 7.3 and 7.4. The artery was secured with nylon suture between two glass cannulas in an organ bath placed on the stage of an inverted microscope (Zeiss Axiovert, Braun; Germany) coupled to a CCD camera (Sony XC-73CE, monochrome, Bangkok; Thailand). After a 15 min equilibration period, intraluminal pressure was reduced to near zero (5 mm Hg) and the arterial segment was exposed to increasing intraluminal pressures in 20 mmHg steps (5, 20, 40, 60, 80, 100, 120, and 140 mm Hg). The passive increase in lumen diameter with increasing distending pressure was considered an indication of segment viability. At each pressure, the artery was allowed to stabilize for three minutes and an image was taken (10× objective). ImageJ software was used to measure internal and external diameters at each intravascular pressure (D_i_, D_e_), and from them, wall thickness (WT) and cross-sectional area (CSA) were calculated. 

To evaluate the degree of arterial stiffness incremental Young’s Elastic Modulus (E_inc_) was used. E_inc_ was obtained by fitting the stress-strain data from each MRA to an exponential curve using the equation: σ = σ_origen_ exp^βε^(1)
Circumferential wall stress (σ) = (P_x_D_i_)/(2WT),(2)
where P is the intraluminal pressure (1 mmHg = 133.4 N/m^2^) and WT is wall thickness at each pressure.
Circumferential wall strain (ε) = (D_i_ − D_0_)/D_0,_(3)
where D_0_ is the diameter at 5 mmHg and D_i_ is the internal diameter at each pressure.

From each stress-strain curve, β values were obtained. For a given level of wall stress (σ), E_inc_ is a measure of stiffness and it is directly proportional to β. Therefore, β values were taken as a measure of the functional stiffness. At the end of the experiment, each MRA segment was pressure-fixed at 80 mm Hg with 4% paraformaldehyde (PFA, in 0.2 M phosphate buffer, pH 7.2–7.4) for 1 h. Thereafter the segment was stored in 4% PFA for confocal microscopy study. 

### 2.4. Confocal Microscopy

Confocal microscopy was used to quantify cell distribution in the adventitia and media, and the internal elastic lamina fenestra organization in the pressure-fixed MRA segments. To assess cellular arrangement the segments were stained with DAPI (1:500 *v*/*v* dilution from stock solution) for 15 min, washed for two periods of 15 min each in saline solution and mounted intact on a slide provided with a small well (to avoid compression), containing Citifluor mounting medium (Life Technologies, Carlsbad, CA, USA).

MRA segments were visualized with a laser scanning confocal microscope (Leica^®^ TCS SP2, Madrid; Spain) with the Excitation = 405 nm/Emission = 410–475 nm wavelengths to visualize cell nuclei (DAPI). Stacks of serial optical sections (1 µm thick) were captured in three randomly chosen regions, starting from the outermost nuclei (adventitia) to the last smooth muscle cell (SMC) visible in the image. At 80 mm Hg, the endothelial cells were not clearly visible for accurate quantification. Quantification was performed with MetaMorph^®^ image analysis software (Universal Image Corporation, San Jose, CA, USA). From each stack of images, we quantified adventitial cell and SMC number (identifiable by the shape and orientation of their nuclei), layer thickness and layer volume (layer thickness × image area). From these data, cell density was calculated in each layer (number of cells/layer volume).

IEL organization was assessed as previously described [21]. IEL was visualized at Excitation = 488 nm/Emission = 500–560 nm, the wavelength at which elastin can be detected by its auto-fluorescence, and stacks of serial optical sections (0.5 µm thick) were captured from three randomly chosen regions with a 20× oil immersion objective at zoom 8, under identical conditions of laser intensity, brightness, and contrast. IEL was reconstructed, and from IEL projections, several measurements were obtained. Fluorescence intensity of elastin (average fluorescent intensity per pixel) was measured in several regions from each projection (minimum of five) and averaged. IEL projections were segmented and binary images were obtained. In the binary images, fenestrae area and number were quantified. 

### 2.5. Zymography

MMP-2 and MMP-9 activity assays were performed by gelatin zymography in plasma samples from 6- and 18-month-old rats. Four μg of proteins were used for MMP-2 and 20 μg of proteins were used for MMP-9. Laemmli solution (0.5M Tris (pH 6.8), 25% glycerol, 20% SDS, and 0.01% bromophenol blue; 1:5 *v*/*v* dilution) was added to plasma samples and loaded to sodium dodecyl sulfate (SDS)-polyacrylamide gels containing 0.1% gelatin. Proteins were separated by electrophoresis at 180 V for 120 min. Thereafter, gels were washed with distilled water, incubated with the enzymatic activation buffer (50 mM Tris-HCl, 6 mM CaCl_2_ and 2.5% Triton X-100) for 1h at room temperature and subsequently with Triton X-100-free activation buffer for 24h at 37 °C. Gels were stained for 10 min with Coomassie Brilliant Blue (BioRad; Hercules, CA, USA) and thereafter washed for 1 min with a solution containing 40% methanol and 10% acetic acid in distilled water. Finally, gels were incubated in a stop solution (10% acetic acid, *v*/*v*) for 24–48 h, and imaged with ChemiDoc XRS+ Imaging System (BioRad, Hercules, CA, USA). Bands were quantified with Image Lab 3.0 software (BioRad, Hercules, CA, USA).

### 2.6. Statistical Analysis

Statistical analysis was performed with GraphPad Prism (version 5, San Diego, CA, USA). The Kolmogorov-Smirnov test was used to check the normal distribution of the variables. Data are expressed as mean ± standard error mean (SEM). Statistical differences between MUN and control rats of each sex were analyzed by Student’s *t* test, 1 or 2-way ANOVA. Statistical significance was established at *p*-value < 0.05. 

## 3. Results

### 3.1. Body Weight

Birth weight was significantly lower in MUN rats compared to Controls, both in males (Control = 7.0 ± 0.1 g; MUN = 4.1 ± 0.2 g; *p*-value < 0.05) and females (Control = 6.8 ± 0.1 g; MUN = 4.4 ± 0.1 g; *p*-value < 0.05). At the age of 6 months, there was no significant difference in body weight between groups either in males (Control = 464.6 ± 14.0 g; MUN = 487.9 ± 13.8 g; *p*-value = 0.25) or in females (Control = 303.3 ± 11.2 g; MUN = 288.9 ± 3.7 g; *p*-value = 0.24). Similarly, at the age of 18 months, no differences were observed between MUN and control rats, either in males (Control = 554.1 ± 5.7 g; MUN = 540.0 ± 6.8 g; *p*-value = 0.16) or females (Control = 356.4 ± 15.1 g; MUN = 363.7 ± 12.7 g; *p*-value = 0.64). 

### 3.2. Hemodynamic Parameters

Under ketamine/medetomidine anesthesia, 6-month-old MUN males exhibited significantly higher SBP and DBP compared to sex-matched Control counterparts. However, no statistical differences were detected between MUN and Control females. Heart rate was not significantly different between MUN and Control rats, either in males or in females. MUN females exhibited significantly lower SBP and DBP values compared to MUN males (Table 1). 

A separate group of rats was used to analyze SBP by tail-cuff plethysmography. The 6-month-old MUN males exhibited significantly higher SBP compared to controls Females tended to have higher blood pressure levels but did not reach statistical significance (*p*-value = 0.19). The 6-month-old MUN females exhibited significantly lower SBP values compared to MUN males (Table 2).

In 18-month-old rats, SBP was significantly higher in both MUN males and females compared to control sex-matched counterparts. No significant difference was detected between MUN males and MUN females (Table 2). These data indicate that male rats exposed to undernutrition develop hypertension in adult age and females develop hypertension only in ageing.

### 3.3. Mesenteric Resistance Artery (MRA) Structure 

MRA internal and external diameters were significantly smaller in MUN male and female rats compared to sex-matched controls at all intraluminal pressures tested (Figure 1). MRA cross sectional area was not different between Control and MUN males (data at 80 mm Hg: Control = 33778 ± 1627 µm^2^; MUN = 31760 ± 1675 µm^2^; *p*-value = 0.20) or females (data at 80 mm Hg: Control = 35197 ± 2500 µm^2^; MUN = 35700 ± 4808 µm^2^; *p*-value = 0.93). These data indicate that exposure to undernutrition induces resistance artery inward eutrophic remodeling, independent of sex.

Confocal microscopy was used to analyze cellular density and thickness in adventitia and medial layers. We did not detect statistical differences in adventitial layer thickness between MUN and control MRA, either in males or in females (Figure 2A). However, cell number was significantly lower in MUN rats with respect to sex-matched controls (Figure 2B). As a consequence, cell density was also smaller in MUN rats, both in males and females (Figure 2C). 

Confocal microscopy showed that media thickness was larger in MUN rats, both in males and females (Figure 3A). The SMC number was not statistically different between groups (Figure 3B) and SMC density was significantly lower in MUN rats compared to controls in both males and females (Figure 3C).

### 3.4. MRA Mechanical Properties

In MUN rats, the stress-strain relationship was shifted to the left and β value derived from the exponential curve was significantly higher compared to Controls, both in males (Figure 4A) and females (Figure 4B). These data suggest that resistance artery remodeling in MUN male and female rats is associated with vascular stiffening.

### 3.5. Internal Elastic Lamina (IEL) 

No significant differences in fluorescence intensity (Figure 5A) or fenestra number (Figure 5B) were observed between MUN and control rats, neither in males nor in females. However, fenestra area was significantly smaller in MUN rats compared to Controls, both in males and in females (Figure 5C). These data indicate IEL from MUN rats remodels, which may contribute to the alteration in mechanical function.

### 3.6. Plasma Metalloprotease (MMP) Activity

In 6-month-old rats, we did not detect differences in MMP-2 activity between MUN and Control rat, either in males or in females. MMP-9 activity was significantly higher in MUN males compared to Control, but there was no statistical difference between MUN and control females (Figure 6A). In 18-month-old rats, MMP-2 activity was not significantly different between MUN and control rats. However, MMP-9 was significantly higher in MUN males and females, with respect to their respective sex-matched controls (Figure 6B). 

## 4. Discussion

This study aimed to assess if exposure to undernutrition during fetal life induces resistance artery remodeling, its relationship with hypertension, and the possible influence of sex. Our main findings are that fetal undernutrition programs MRA inward eutrophic remodeling and stiffening. These structural and mechanical alterations are present in adult MUN rats from both sexes. Since males and females exhibit different levels of blood pressure at this age, these data suggest that remodeling may occur prior to the development of hypertension, as we have previously demonstrated in conduit vessels from 21-day-old MUN rats. Resistance artery remodeling may be a contributing factor to the maintenance of hypertension in MUN males and later development of high blood pressure in aged MUN females. Our data also suggest that plasma MMP activation may be the result of hypertension affecting all the vasculature. However, we cannot discard that alterations in local MMP expression or activity may contribute to the observed remodeling. We suggest that resistance artery remodeling can contribute to cardiovascular organ damage associated with low birth weight, particularly in ageing, when the protection of females due to estrogen loss is blunted.

### 4.1. Hypertension Development in Males and Females

CVDs are still the leading cause of death due to non-communicable diseases, despite efforts to control lifestyle factors. It is now well accepted that fetal stress and associated inadequate growth contributes to the development of hypertension, which is a risk factor for coronary heart disease and stroke [2,3]. The majority of studies in animal models of fetal stress, show that females have a certain degree of protection [22], also confirmed in our previous reports in the MUN rat model of fetal programming [8,17]. However, some reports demonstrate that females exposed to several developmental insults during fetal life develop milder forms of hypertension [16], or develop hypertension at later stages of life compared to males, possibly related to the amplifying effects of ageing [2]. In our previous study in 20-month-old females, we found that they did not develop hypertension at this age [19]. However, we performed the experiments under ketamine/medetomidine anesthesia, which can exert cardiovascular effects, affecting sympathetic activity. Alterations in the sympathetic nervous system have been described in animal models of fetal programming [22,23]. Therefore, to avoid the influence of the anesthesia, we performed experiments in awake rats by tail-cuff plethysmography. The 6-month-old MUN females did not exhibit significant differences in blood pressure. However, 18-month-old MUN females were hypertensive. Taken together, the present results and our previous findings in anesthetized rats [19], we suggest that MUN females develop hypertension in ageing and that anesthesia masks blood pressure elevation. It is possible that this effect is related to a sympathetic over activation in MUN females. In spontaneously hypertensive rats (SHR), it has been demonstrated that development of hypertension in females, which also takes place in old age, is related to sympathetic nervous system activation [24]. If the same mechanism take place in females exposed to fetal stress deserves further analysis. 

### 4.2. Resistance Artery Remodeling and Hypertension

The association between inadequate fetal growth and CVD has been proposed to take place through two main pathways: cardiovascular remodeling and metabolic programming. In our rat model of low birth weight, we found that adult rats exposed to undernutrition during fetal life exhibit remodeling of third order branches of the mesenteric artery (MRA), which is an example of resistance artery in the rat. The structural alteration was found in both MUN males and females, despite the fact that at the age of 6 months, only males developed hypertension. This suggests that the process of vascular remodeling in rats exposed to undernutrition may be independent on the level of blood pressure. It is likely that vascular structural alterations are initiated early in life. This is supported by data from our previous study in rat aorta, showing that both MUN males and females exhibit remodeling by the age of 21 days, when males have not developed high blood pressure yet [8]. Early signs of remodeling have also been observed in the aorta from newborn rats exposed to undernutrition [13], in sheep fetuses exposed to placental insufficiency [25], and in fetus from rats exposed to chronic hypoxia [26]. Besides, human studies have also revealed increased aortic intima-media thickness in children with fetal growth restriction [27,28]. Taken together, these data support the idea that vascular remodeling can be programmed in utero, irrespective of the type of fetal insult, and is not the consequence of high blood pressure. 

Resistance artery remodeling is well known to be a contributing factor for hypertension development, through the increase in peripheral resistance. We were surprised to find that, despite the presence of MRA inward eutrophic remodeling, MUN females did not develop hypertension. This finding suggest that females may have mechanisms, which may counteract increased peripheral resistance. Sex differences in fetal programming of hypertension have been attributed to sex hormones, which exert actions on several blood pressure regulatory systems [29]. Among others, it has been reported that females exhibit better endothelial function [30,31], reduced responses to the renin-angiotensin system—which is downregulated by estrogens and amplified by testosterone—or better oxidative balance [17,18]. It is also possible that a lower MRA myogenic tone may contribute to maintain normal blood pressure level in females, despite the observed structural alteration. Resistance vessels exhibit myogenic tone, i.e., they constrict in response to pressure increases. In this study, we performed the experiments in zero calcium conditions and myogenic tone was not evaluated. In addition to the role of myogenic tone in blood pressure, it can also be a factor in the remodeling process. Increased myogenic tone maintained for a long time, leads to reorganization of cytoskeleton, which may participate in the initial phases of vascular remodeling [32]. Myogenic tone has been studied in cerebral vessels from rats exposed to fetal undernutrition. However, in this study, a depressed, rather than enhanced myogenic tone was found, despite enhanced stiffness and increased wall thickness [33]. It is possible that cerebral and systemic arteries respond differently to pressure overload and myogenic constriction. This aspect deserves further attention. 

### 4.3. Characteristics of MRA Remodeling 

The vascular structural alteration observed in MUN MRA was characterized by a reduced lumen size with no differences in cross sectional area, and, therefore, can be classified as inward eutrophic remodeling [10]. This type of remodeling is well known to be a characteristic feature of essential hypertension both in humans [34,35] and rats [21,36]. With the aid of confocal microscopy, we demonstrated that all the layers of the vascular wall—adventitia, media, and intima—were affected. Adventitia exhibited a reduction in the number of cells without alterations in thickness, and subsequently, cell density was reduced. Medial layer increased in thickness and cell density was also reduced. These findings suggest that either cells are hypertrophied, or they are substituted by ECM. SMC exhibit high plasticity, and phenotypic switch towards a secretory non-contractile phenotype can lead to ECM accumulation. Loss of contractile proteins suggesting this phenotypic switch have been found in MRA from rats exposed to fetal undernutrition and in a rat model of fetal programming induced by uterine artery ligation, although this was observed only in male offspring [37]. Previous reports in animals exposed to stress factors during intrauterine life have demonstrated alterations in several ECM proteins, including elastin, collagen, and glycosaminoglycans [13,25]. ECM is a key determinant of vascular passive mechanical properties and alteration in their organization or content is associated with stiffness [38]. We found that MRA remodeling was accompanied by increased stiffness, as shown by the leftward shift of the stress-strain relationship and higher β values derived from the exponential curves. These mechanical alterations are similar to those found in rat models of essential hypertension, such as the SHR [21,36]. Increased arterial stiffness has also been described in humans with low birth weight, even at a young age [39,40]. Collagen has been regarded as the key protein responsible for vascular stiffness. However, in our previous study, we did not find differences in collagen content in aorta from six month-old MUN rats [8]. In addition to collagen, other ECM elements, such as fibronectin and elastic fibers, play a role [38]. In MRA from SHR we have previously demonstrated that elastin distribution, and fenestra size are also key determinants of vascular mechanical function, and smaller fenestra are associated with vessel narrowing and stiffening [21]. Elastogenesis is physiologically produce only during fetal period and early postnatal years, and thereafter, elastic fibers are only exposed to fatigue and degradation [38]. Alterations in elastin distribution, with a decrease in inter-elastin laminae, have been observed at birth in the aorta from rats exposed to maternal undernutrition [13] and in aorta from ovine fetus exposed to hypoxia [25]. However, in pulmonary vessels from rats exposed to undernutrition there is evidence of increased elastin deposition at weaning, which persist until adult age [41]. These data suggest that a defective elastin deposition due to disturbed fetal environment, followed by abnormal elastin remodeling along lifespan, may contribute to a mechanical dysfunction. 

### 4.4. Role of MMPs

MMPs are the key enzymes responsible for degradation of various ECM proteins in the vascular wall, including collagen, elastin, glycoproteins, and proteoglycans, and MMP dysregulation is associated with pathological vascular remodeling [12]. In rats exposed to fetal undernutrition Khorram et al. found an increase in MMP-9 expression in rat aorta already at the age of one day [13] and small-for-gestational-age children also exhibited elevated plasma levels of MMP-2 and MMP-9 [14]. These data suggest that ECM degradation by MMPs is initiated early in life. The association between MMP activation and remodeling has also been found in animal models of hypertension. In 2K-1C hypertensive rats, in situ zymography demonstrated increased MMP-2, MMP-9, and MMP-14 activity in the aorta in association with remodeling [42]. The activation of MMPs and degradation of collagen and elastin fibers favor the unleashing of SMCs from ECM proteins, and participate in the SMC shift to a hypertrophic and proliferative phenotype [15]. However, despite evidence of MRA remodeling in MUN males and females, our data showed an elevation of MMP-9 activity only in males. We suggest that increased plasma MMP-9 activity may be related to hypertension. In fact, a correlation between MMP levels and hypertension has been previously described [12]. Higher serum concentrations of several MMPs have been found in hypertensive patients and in several animal models of hypertension; there is also evidence that MMP levels are reduced upon antihypertensive treatments [15]. We did not evaluate in situ zymography, which is a limitation of our study. Therefore, it is possible that local MMP expression or activation may occur in MUN MRA arteries in association with the observed structural alterations. 

The majority of studies regarding vascular remodeling in fetal programming have focused on elastic arteries. The present data evidence that resistance vessels also undergo structural and mechanical alterations. Small artery remodeling is not only a key determinant of peripheral resistance and hypertension, but also a contributing factor to adverse cardiovascular events [10,11,43] and may be the first manifestation of target organ damage [44]. MRA remodeling is likely to reflect structural alterations in other vascular beds, such as coronary or brain arteries. We have evidence of intramyocardial artery remodeling in MUN rats (data not shown) and the development of cardiac dysfunction in aged MUN males and females [19]. Therefore, resistance artery remodeling may play a role in the development of CVDs associated with suboptimal fetal growth, and evaluation of resistance artery structure and function in individuals born small could be of value, particularly in young subjects, in order to prevent future development of CVDs in this population.

## 5. Conclusions


Fetal undernutrition induces MRA inward eutrophic remodeling in males and females, independent of blood pressure level.MRA remodeling in MUN rats is associated with cellular and ECM alterations and stiffening.MUN females develop hypertension in ageing, and MRA remodeling may be a contributing factor.Increased plasma MMP-9 activity is associated with hypertension.Resistance artery remodeling induced by fetal undernutrition may contribute to organ damage and development of CVD.


## Figures and Tables

**Figure 1 biomedicines-08-00424-f001:**
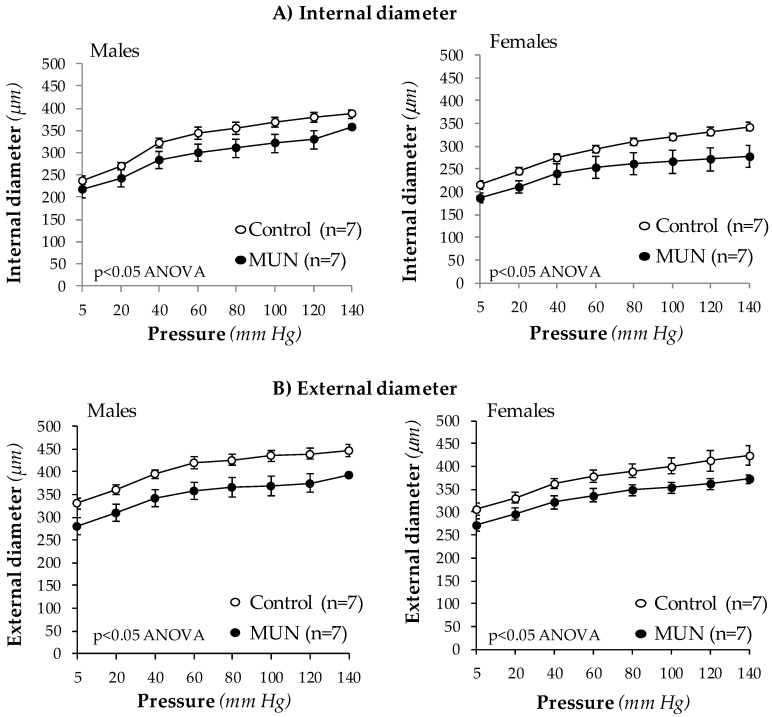
Internal (**A**) and external diameters (**B**) in MRA from 6-month-old rats. MUN, maternal undernutrition. Data show mean ± SEM; n, sample size. Statistical differences analyzed by two-way ANOVA.

**Figure 2 biomedicines-08-00424-f002:**
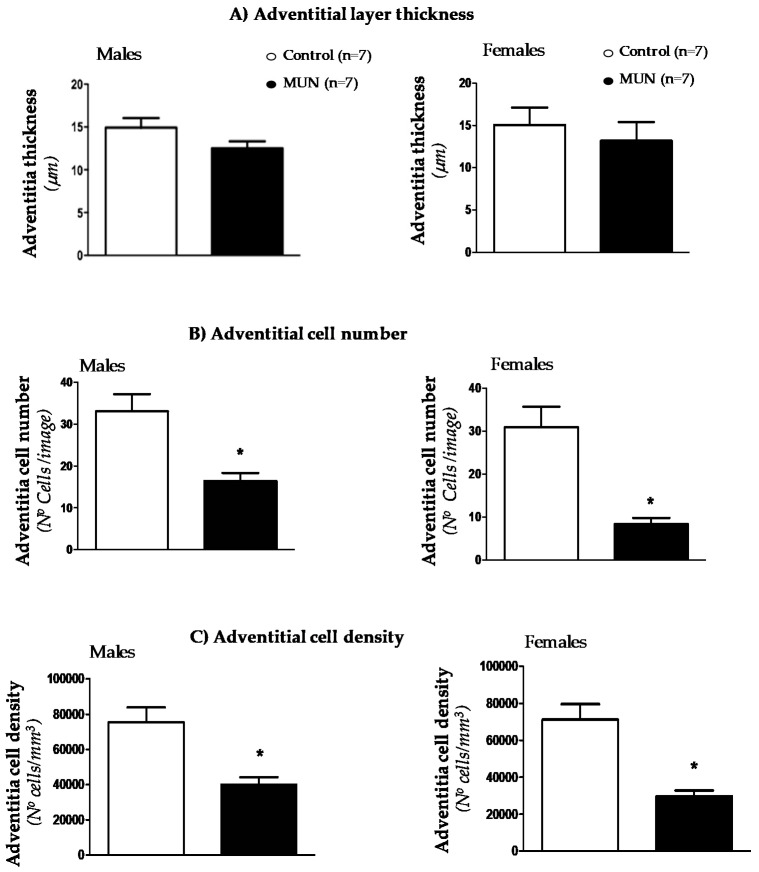
Thickness (**A**), cell number (**B**) and cell density (**C**) in MRA adventitial layer from 6-month-old MUN and Control rats. MUN, maternal undernutrition. Data show mean ± SEM; n, sample size. Student’s *t* test; * *p*-value < 0.05 compared to sex matched controls.

**Figure 3 biomedicines-08-00424-f003:**
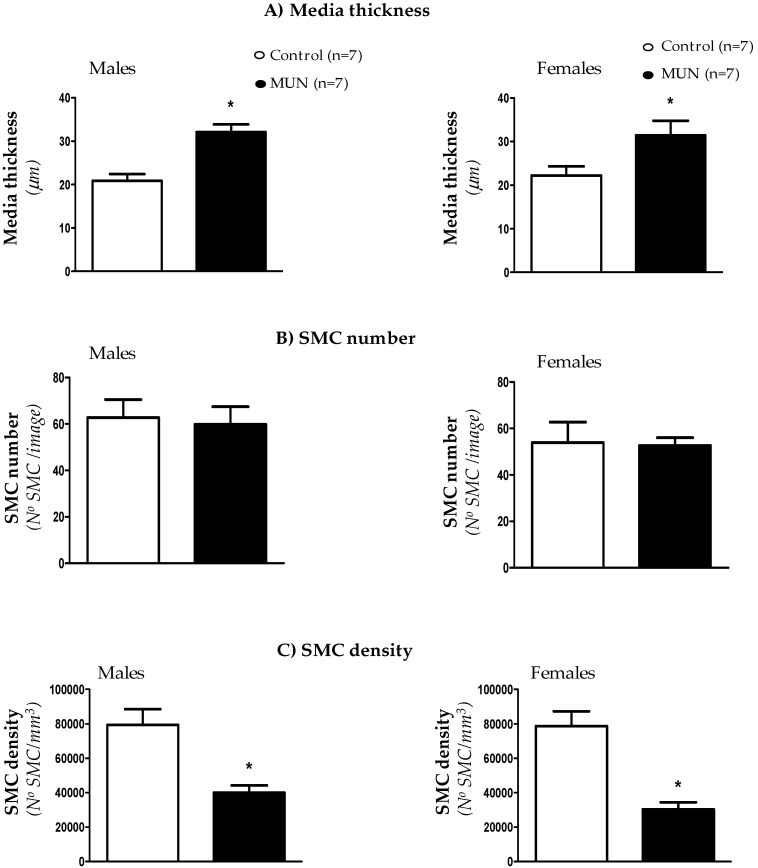
Thickness (**A**), SMC number (**B**) and SMC density (**C**) in MRA medial layer from 6-month-old MUN and control rats. SMC, Smooth Muscle Cell; MUN, maternal undernutrition. Data show mean ± SEM; n, sample size. Student’s *t* test; * *p*-value < 0.05 compared to sex matched controls.

**Figure 4 biomedicines-08-00424-f004:**
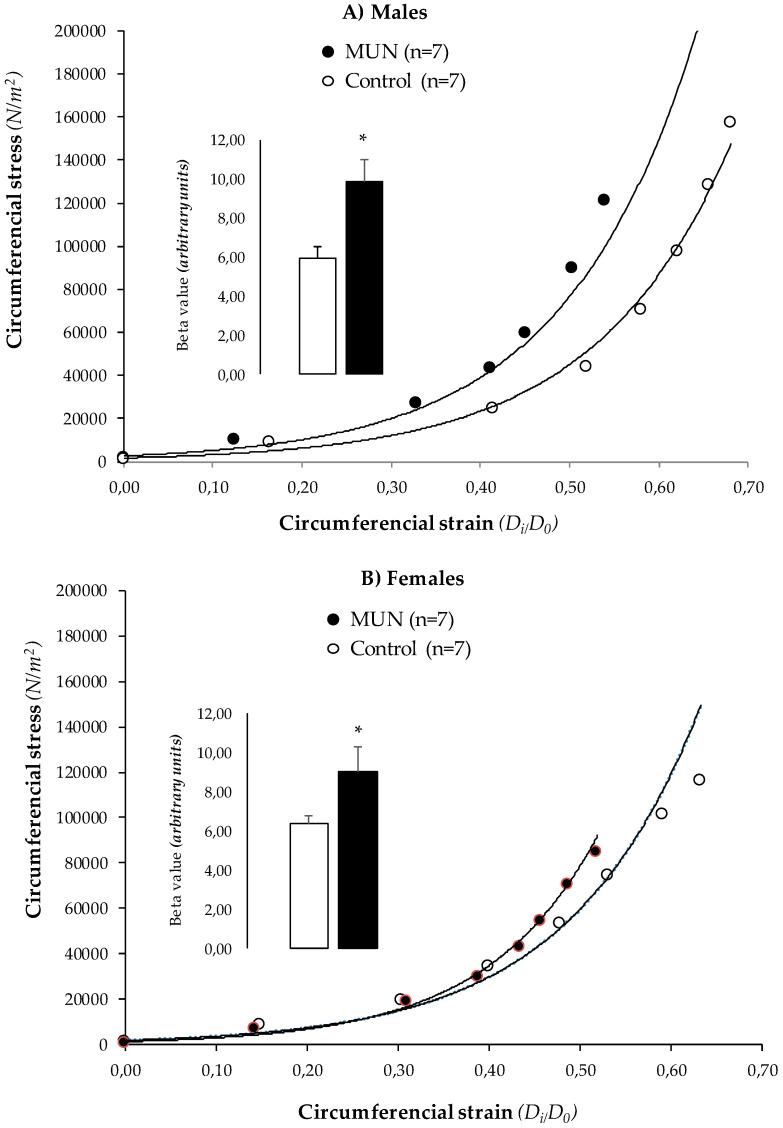
Stress-strain relationship in MRA from 6-month-old MUN and Control rats. (**A**) Males and (**B**) females. Inset shows beta values derived from the exponential curves. MUN, maternal undernutrition. Data show mean ± SEM; n, sample size. Student’s *t* test; * *p*-value < 0.05 compared to sex matched controls.

**Figure 5 biomedicines-08-00424-f005:**
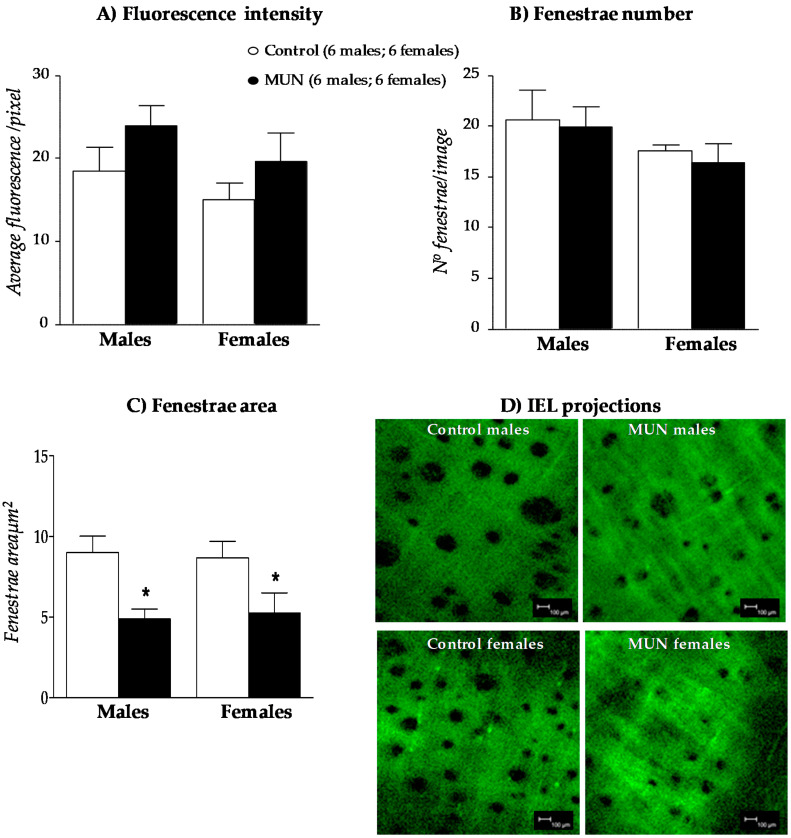
Internal elastic lamina in MRA from MUN and control male and female rats. Fluorescence intensity (**A**), fenestra number (**B**), fenestra area (**C**), and representative reconstructions (**D**). Reconstructions are projections of stacks of images obtained with confocal microscopy (scale bar = 100 μm). MUN, maternal undernutrition. Data show mean ± SEM. Sample size is shown between parentheses. Student’s *t* test; * *p*-value < 0.05 compared to sex matched controls.

**Figure 6 biomedicines-08-00424-f006:**
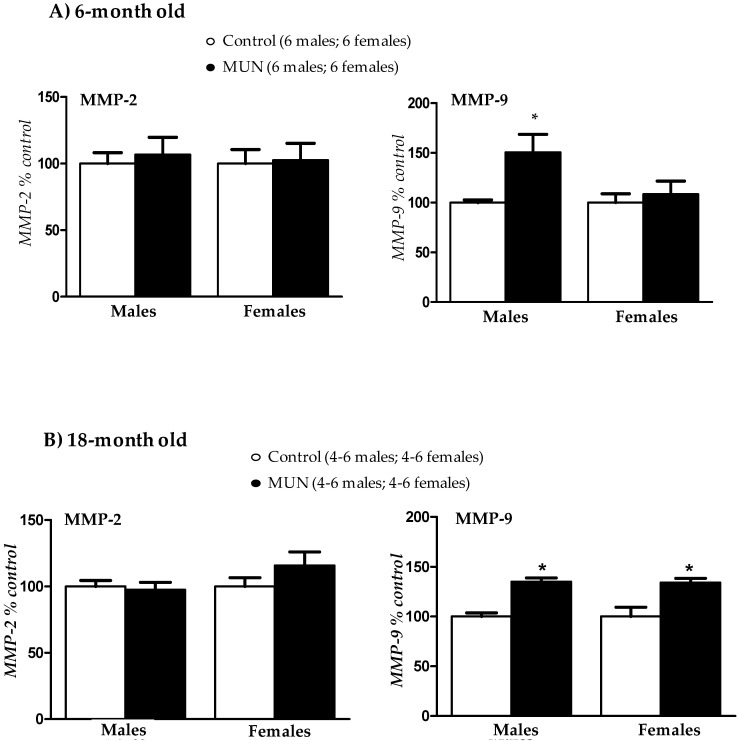
Plasma metalloprotease-2 and 9 activity assessed by zymography in 6-month-old (**A**) and and 18-month-old rats (**B**). MUN, maternal undernutrition, MMP, metalloprotease. Data show mean ± SEM; Sample size is shown between parentheses. Student’s *t* test; * *p*-value < 0.05 compared to sex matched control.

**Table 1 biomedicines-08-00424-t001:** Hemodynamic parameters in anesthetized maternal undernutrition (MUN) and Control 6-month-old rats.

	Males		Females	
	Control (n = 7)	MUN (n = 7)	Control (n = 7)	MUN (n = 7)
SBP (mm Hg)	125.6 ± 5.1	157.4 ± 2.9 *	134.2 ± 3.4	135.5 ± 4.0 ^#^
DBP (mm Hg)	68.7 ± 4.1	95.8 ± 4.3 *	64.2 ± 3.6	76.8 ± 4.1 ^#^
HR (beats/min)	258.4 ± 8.4	258.3 ± 10.3	250.1 ± 9.6	234.2 ± 9.7

Data show mean ± SEM; n, sample size. SBP, systolic blood pressure; DBP, diastolic blood pressure; HR, heart rate; MUN, maternal undernutrition. * *p*-value < 0.05 compared to sex and age-matched controls; ^#^
*p* < 0.05 compared to males of the same experimental group (1-way ANOVA).

**Table 2 biomedicines-08-00424-t002:** Systolic blood pressure (mm Hg) in awake MUN and Control 6- and 18-month-old rats

	Males		Females	
	Control	MUN	Control	MUN
6-month-old	147.7 ± 3.1 (n = 9)	163.1 ± 2.8 * (n = 10)	136.9 ± 2.3 (n = 9)	145.0 ± 4.5 ^#^ (n = 10)
18-month-old	137.7 ± 3.2 (n = 10)	162.2 ± 3.2 * (n = 9)	131.0 ± 4.7 * (n = 7)	152.7 ± 5.2 * (n = 7)

Data show mean ± SEM; n, sample size; MUN, maternal undernutrition. * *p*-value < 0.05 compared to sex and age-matched controls; ^#^
*p* < 0.05 compared to males of the same experimental group (1-way ANOVA).

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
