# Peer review of "Fetal Undernutrition Induces Resistance Artery Remodeling and Stiffness in Male and Female Rats Independent of Hypertension"

_biomedicines, 2020, doi:10.3390/biomedicines8100424_

Round 1
Reviewer 1 Report
General Comments:
In this manuscript, the authors examine effects of maternal undernutrition (MUN) on resistance artery remodeling in offspring and its relationship with development of hypertension. This paper indicates that MUN males exhibited higher blood pressure compared to control, whereas no noticeable difference were observed in female offspring. The authors also discovered that both male and female MUN offspring exhibited decreased internal and external diameters and a left-ward shift in the stress-strain curve when compared to controls. Additionally, the authors observed an overall decrease in cell density in the adventitial and medial layer, in addition to a decrease in the IEL fenestra area in MUN males and females. Finally, the authors demonstrated that MMP-9 activity was significantly greater in MUN males but not MUN females. Overall, the experiments appear to be well thought out and provides some insight into how insufficient nutrition could contribute to resistance vascular arterial remodeling and the future development of CVD. However, there are some major concerns regarding the experimental details of this study that reduce my enthusiasm for this manuscript. In addition, this manuscript needs to be substantially revised for proper use of the English language.
Specific Comments:
- The authors only examined the passive diameter of the mesenteric resistance arteries. I feel that they missed out on the opportunity to explore how maternal undernutrition effects myogenic tone. Furthermore, the author did not indicate how they determined viability before or after vessels were mounted to the pressure myograph.
Minor Comments:
- This manuscript needs to be substantially revised for proper use of the English language.
- Page 10, line 32-33: You wrote down that MMP-2 activity is increased in MUN males, but your data would suggest that MMP-9 is increased. I suspect that this was a typo. Please correct.
- Page 11, line 1: It should be MMP-2 not MMP-9. Please correct.
Reviewer 2 Report
This study assesses the role of maternal undernutrition (MUN) on blood pressure and arterial structure. While providing new insight into the effects of MUN in resistance arteries, the present manuscript is mostly observational in nature and provides little mechanistic insight into the how the observed structural changes occur and why males but not females exposed to undernutrition during fetal development develop hypertension. The relationship between these factors remains ambiguous, and the authors speculate to the role of changes in extracellular matrix without providing any direct evidence. General and specific comments are included below:
Broad Comments:
- The present study addresses the role of MUN in resistance arteries, not large conduit vessels, an area which had been relatively unaddressed.
- The authors adequately highlight limitations of methods, including effects of anesthesia on blood pressure measures.
- The present study provides limited mechanistic insight into responses. This is somewhat addressed with MMP zymography, but results do not suggest them as the primary mediator of the alterations in vessel wall structure/proposed extracellular matrix deposition due to lower cell density. Whether and how extracellular matrix is altered is not clearly addressed. Furthermore, the relationship between MMP differences in males but not females appears to only correlate to blood pressure, and not arterial remodeling despite their role in extracellular matrix control. At present, this relationship is poorly defined within the manuscript.
- The authors seem to waver on whether the arterial structure changes could be a factor in the hypertension observed in males (2nd discussion paragraph females have compensatory mechs, 3rd paragraph no relationship). While the reviewer agrees this is unclear at the present stage of research, writing should be revised to more clearly identify this phenomenon as something that requires further investigation.
Specific Comments:
Major concerns:
- The authors highlight the role of fetal undernutrition in hypertension and coronary disease in the introduction. However, the rationale for their focus on resistance changes in mesenteric arteries is not fully developed (i.e. Why was this vascular bed chosen over others that have a clearer role in undernutrition such as cerebral or coronary resistance arteries?).
- Why were arteries exposed to hyperoxic conditions (95% O2) for pressure myography as opposed to standard normoxic (~21% O2) conditions?
- The authors demonstrate differences in arterial diameter due to malnutrition in response to increasing pressure steps under passive (Ca2+ free conditions). However, the functional significance of this observation is limited by whether this phenomenon contributes to in vivo changes in pressure. Performing pressure response curves under both Ca2+ replete and deplete conditions could determine whether there is a compensatory change in myogenic (pressure-dependent) tone between control and MUN rats. These findings would more clearly demonstrate the relationship between pressure/hypertension and arterial wall remodeling, particularly with regards to how both males and females demonstrate remodeling but only males develop hypertension.
- As the authors demonstrate inward remodeling, it is surprising that the intimal (endothelial) thickness, cell #, and density are not quantified in a manner similar to adventitial and smooth muscle cells (Figures 2 and 3). These data should be included to assess any potential changes in this cell layer that contribute to intimal remodeling.
- In Figure 5, the authors demonstrate differences in fenestrae area without changes in number suggesting a reduction in size. A quantification of mean fenestrae size would strengthen this observation.
- The rationale for studying MMPs 2 and 9 is not clearly indicated in the introduction, methods, or results and should be added to facilitate understanding of experiments to the reader (i.e. potential to induce changes in elastic lamina) and not only presented in the discussion. Additionally, are plasma levels truly indicative of what occurs in the vessel wall? Zymography on vessel wall homogenates would be a better indicator of what occurs within the artery.
- The conclusions are not fully corroborated by the manuscript. The 1st point suggests structural changes play no role in blood pressure; however, the 3rd point suggests a potential role. In the 2nd point, ECM alterations are not truly investigated within the present study. In the 4th point, the reason for increased MMP-2 activity is not addressed in the manuscript, and suggesting it is induced by hypertension is just speculation. Additional experimentation would be required to make this claim.
Minor concerns:
- Page 2, line 23: “seems” should be “seem”. Line 25:” evidenced” should be “evidence”.
- Page 3, line 18: “store” should be “stored”. Following sentence should read, “Quantification of heart rate (HR; beats/min) and diastolic and systolic blood pressure (DBP, SBP, mmHg) was performed…”
- Page 3, line 38: It would be useful to define variable “e” in the equation for the reader.
- Confocal microscopy: Objective magnification for nuclear counts should be included.
- Section 2.5 title: “Zimography” should be “Zymography”.
- Table 1 legend: “media” should be replaced with “mean values” or something similar.
Reviewer 3 Report
the manuscript is the result of a fruitful experimental work; it has been well structured and the methodological part was well conducted. The results, accompanied by adequate and explanatory tables with images, were well expressed.
In order to make the manuscript more complete, however, the authors should, in the Discussion session, well illustrate the pathophysiological mechanism underlying the hypertensive process and, if present in the literature, illustrate the collateral role of other MMP-s in the pathogenesis of the samet.
Round 2
Reviewer 2 Report
This study assesses the role of maternal undernutrition (MUN) on blood pressure and arterial structure. The revised manuscript has greatly improved the rationale for experiments as well as assisting the reader with an understanding of how responses may differ between sexes. These findings provide new insight into the effects of MUN in resistance arteries and the additional data in aged rats helps to explain sex differences. The conclusions have also been clarified to make the relationship between factors less ambiguous. While substantial improvements to the manuscript have been made in this version of the manuscript, there are a few remaining concerns to address:
Broad Comments:
- The present study addresses the role of MUN in resistance arteries, not large conduit vessels, an area which had been relatively unaddressed.
- The authors adequately highlight limitations of methods, including effects of anesthesia on blood pressure measures.
- Changes to the manuscript have improved rationale, data clarity, and conclusions.
Specific Comments:
Major concerns:
- In table 2, it would improve clarity if there were different significant difference symbols for age and sex effects.
Minor concerns:
- Page 2, Line 73: “Individual" should be “Individuals”. Line 91: “observed” might be a better word choice than “evidenced”. "Evidenced", used throughout the manuscript, is not incorrect, but also not a regularly used English term.
- Page 4, line 150: Please revise to “rats could be accustomed to the challenge/procedure” for clarity. As written the idea seem incomplete.
- Section 3.3: If would be beneficial to combine the first two paragraphs as the topics relate to each other and the first paragraph is currently limited to 1 sentence.
Author Response
ANSWERS TO REVIEWER 2
Specific Comments:
Major concerns:
- In table 2, it would improve clarity if there were different significant difference symbols for age and sex effects.
RESPONSE: We have included the analysis of the influence of sex in table 2 and in table 1. We have also explained the results the text (lines 238-239 and 246-250).
Minor concerns:
- Page 2, Line 73: “Individual" should be “Individuals”. Line 91: “observed” might be a better word choice than “evidenced”. "Evidenced", used throughout the manuscript, is not incorrect, but also not a regularly used English term.
RESPONSE. We have corrected the error. We have also changed the word “evidenced” to “observed/demonstrated/showed” in other parts of the text; we agree that this word was very repetitive.
- Page 4, line 150: Please revise to “rats could be accustomed to the challenge/procedure” for clarity. As written the idea seem incomplete.
RESPONSE. We have completed the sentence and hope the idea is now clearer.
- Section 3.3: If would be beneficial to combine the first two paragraphs as the topics relate to each other and the first paragraph is currently limited to 1 sentence.
RESPONSE. We have combined the paragraphs, as suggested.